# Global eye health and the sustainable development goals: protocol for a scoping review

Justine H Zhang ,[1,2] Jacqueline Ramke ,[1,3] Nyawira Mwangi,[1,4] João Furtado,[5] Sumrana Yasmin,[6] Covadonga Bascaran,[1] Cynthia Ogundo,[1,7] Catherine Jan,[8] Iris Gordon,[1] Nathan Congdon,[9] Matthew J Burton[1,10]

For numbered affiliations see end of article.

**Correspondence to**
Dr Justine H Zhang;
Justine.zhang@lshtm.ac.uk

## ABSTRACT

**Introduction** In 2015, most governments of the world committed to achieving 17 sustainable development goals (SDGs) by the year 2030. Efforts to improve eye health contribute to the advancement of several SDGs, including those not exclusively health-related. This scoping review will summarise the nature and extent of the published literature that demonstrates a link between improved eye health and advancement of the SDGs.

**Methods and analysis** Searches will be conducted in MEDLINE, Embase and Global Health for published, peer-reviewed manuscripts, with no time period, language or geographic limits. All intervention and observational studies will be included if they report a link between a change in eye health and (1) an outcome related to one of the SDGs or (2) an element on a pathway between eye health and an SDG (eg, productivity). Two investigators will independently screen titles and abstracts, followed by full-text screening of potentially relevant articles. Reference lists of all included articles will be examined to identify further potentially relevant studies. Conflicts between the two independent investigators will be discussed and resolved with a third investigator. For included articles, data regarding publication characteristics, study details and SDG-related outcomes will be extracted. Results will be synthesised by mapping the extracted data to a logic model, which will be refined through an iterative process during data synthesis.

**Ethics and dissemination** As this scoping review will only include published data, ethics approval will not be sought. The findings of the review will be published in an open-access, peer-reviewed journal. A summary of the results will be developed for website posting, stakeholder meetings and inclusion in the ongoing Lancet Global Health Commission on Global Eye Health.

## INTRODUCTION

In 2015, at the United Nations (UN) Sustainable Development Summit most governments around the world committed to the sustainable development goals (SDGs). The 17 SDGs have 169 targets and 232 indicators that UN member states aim to achieve by 2030.[1] The SDGs build on the millennium development goals and UN member states are expected to use the SDGs as a framework to guide

| Strengths and limitations of this study |
| --- |
| ► This is the first review to determine the nature and extent of published literature on how improvements in eye health contribute to the advancement of the sustainable development goals (SDGs). |
| ► The review will comprehensively assess published peer-reviewed manuscripts, with no time period, language or geographic restrictions. |
| ► A potential limitation might be the paucity of published literature on how eye health contributes to some of the SDGs. |
| ► Another potential limitation is that the complexity of pathways between eye health and the SDGs is unlikely to be fully appreciated from the published literature. |

the development of national and international policies. The SDGs are broad and far-reaching, with a vision for global change that encompasses the advancement of economic, health, education, equality, social and environmental issues. All of the SDGs are inter-linked, for example, improving health goes hand-in-hand with ending poverty, reduction of inequalities and strengthening the economy.

We hypothesise that improved eye health contributes to the advancement of multiple SDGs. The aim of this review is to summarise the nature and extent of the published literature that demonstrates this link between improved eye health and the advancement of the SDGs. We consider improved eye health to include the full range of promotion, prevention, treatment and rehabilitation strategies.[2] Improving eye health is not only about improving sight, but also, reducing disability, morbidity (eg, pain) and improving well-being. We chose to undertake a scoping review rather than an alternative evidence synthesis approach because we wished to

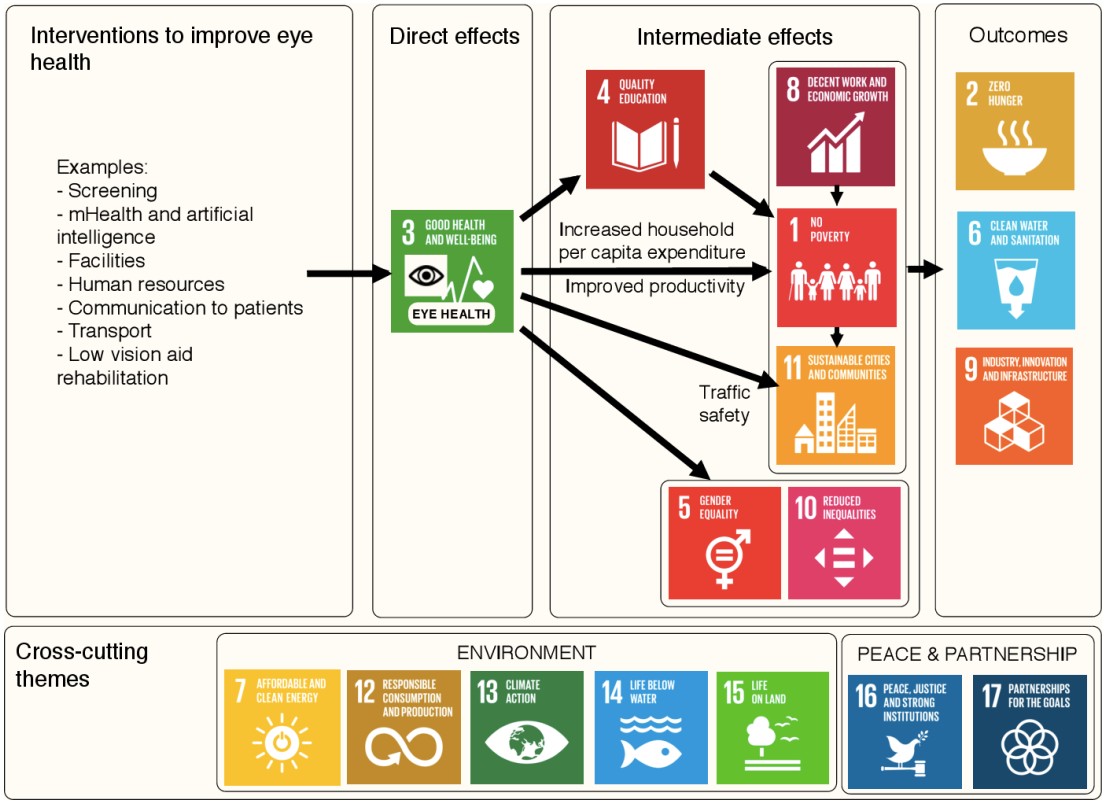

**Figure 1**   Logic model outlining examples of pathways by which improved eye health may contribute to the achievement of the sustainable development goals.[7] (Arrows represent directionality, and not necessarily causality; the links between eye health and 'SDG 3 Good Health and Wellbeing' are considered in a complimentary review.)

identify and map the available evidence, which we anticipate will be heterogeneous.[3]

To guide our review, we developed a model to conceptualise how improved eye health leads to realisation of a range of SDGs (figure 1).[4] To develop this model we asked Commissioners of the ongoing Lancet Global Health Commission on Global Eye Health to review each of the 169 SDG targets and outline the links (hypothesised or established) between any of these targets and eye health.[1 5] These links were reviewed, and an initial logic model drafted and iteratively refined with input by the authorship group.[4] In the model (figure 1), we have depicted SDG 3 as improved eye health, since eye health comes under the umbrella of 'SDG 3 Health and Wellbeing'. We considered the SDGs relating to the environment, peace and partnership as cross-cutting themes.

We recognise that eye health has consequences for other health and wellbeing outcomes, but in this scoping review we aim to identify the broader societal implications linked to eye health. Alongside this review, and also within the Lancet Global Health Commission on Global Eye Health, we are undertaking a complementary review of the intersections between eye health and other health and wellbeing outcomes that will be published separately.

We also recognise that some of the relationships between eye health and the SDGs might be bi-directional. For example, reduced hunger (SDG 2) reduces malnutrition-related eye disease (and thus improves eye health), but conversely, improved eye health reduces poverty and thus reduces hunger (SDG 2). However, for the purposes of this review, we are focused on the contribution that improved eye health can make to other SDGs.

## Objectives/scoping review questions
We aim to answer the following questions:
1. What is the nature and extent of the published evidence that improving eye health contributes to advancement of the SDGs?
2. What are the main pathways by which improving eye health leads to advancement of the SDGs?

## METHODS AND ANALYSIS
### Protocol and registration
This protocol is reported according to the relevant sections of the Preferred Reporting Items for Systematic Reviews and Meta-Analyses Extension for Scoping Reviews (PRISMA-ScR) Checklist (online supplementary appendix 1). The protocol has been registered prospectively with Open Science Framework: https://osf.io/gu4z6/.

### Eligibility criteria
Type of study: primary research studies or meta-analyses only. All intervention and observational studies will be included if they report a link between a change in eye health and: (1) an outcome related to one of the SDGs

(see online supplementary appendix 2 for list of indicative outcomes) or (2) an element on a pathway between eye health and an SDG (eg, productivity; see online supplementary appendix 2 for list of indicative pathway elements). For systematic reviews without a meta-analysis, we will examine the reference list for potentially relevant studies but the actual systematic review will not be included.

Time period: all time periods.

Setting: studies can be from any world region.

Language: manuscripts in all languages will be included for screening. Best efforts will be made to translate any foreign language publications. For any publication where translation is not possible, we will refer to English language versions of their abstract (if available).

Publication status: published peer-reviewed primary research studies or meta-analyses only. As this scoping review is concerned with identifying the extent of evidence in published literature, grey literature will not be searched.

## Search

We will search MEDLINE, Embase and Global Health using a search strategy developed by an Information Specialist from Cochrane Eyes and Vision (IG). The search was constructed using a set of terms describing eye health, and they were combined with a set of terms describing each of the SDGs (except for SDG 3). Our search excluded the following study designs by using 'stop words': animal, laboratory, case reports and case series. Our MEDLINE search strategy is included in online supplementary appendix 3. We will download and de-duplicate the results in EndNote, and then export the results into Covidence (Veritas Health Innovation, Melbourne, Australia. Available at www.covidence.org) for screening. We will examine reference lists of all included articles to identify further potentially relevant studies. Following the selection process, field experts will be provided with a list of the included studies and requested to identify further potentially relevant studies.

## Selection of sources of evidence

All titles and abstracts will be screened by two investigators independently using Covidence systematic review software. Subsequently, full texts will be assessed by two investigators independently to establish eligibility for inclusion into the study, and reasons for exclusion will be assigned by each investigator. Any conflicts will be discussed and resolved with a third investigator. A PRISMA flow diagram will be completed to summarise the study selection process.

## Data charting process

Data charting forms will be developed in Excel and tested by all investigators on two studies each prior to use, based on the data items listed below. Data charting of included studies will be performed by two investigators independently. We anticipate a broad scope of included studies, so data charting will be an iterative process throughout the review and the data charting form will be amended as required. We plan to contact study authors in the case of unclear information and will make up to three attempts by email.

## Data items

► Publication characteristics:
  – Authors.
  – Date of publication.
  – Journal.
  – Country of study.
  – Source of funding and role of funder.
  – Type of study: (1a) intervention, randomised; (1b) intervention, non-randomised; (1c) intervention, model; (2a) observational, cohort, analytical; (2b) observational, cohort, descriptive; (2c) observational, case-control; (2d) observational, cross-sectional, analytical; (2e) observational, cross-sectional, descriptive; (2f) observational, ecological; (3) other—please state.
► Study details:
  – Exposure(s)/intervention(s).
  – Outcome(s).
  – Effect estimate(s).
► SDG-related characteristics:
  – Identify relevant SDG (or multiple relevant SDGs).
  – Specify outcome(s) related to one or more of the SDGs (and targets if possible).
  – Describe the link between eye health and the SDG(s) (and the pathway if available).
  – Map to logic model (figure 1).
  – Identify any links with the environment SDGs (SDG 7, 12, 13, 14 and 15).
  – Identify any links with peace and partnership SDGs (SDG 16 and 17).

## Synthesis of results

Following data charting, results will be synthesised by mapping the retrieved evidence to an Eye Health—SDG logic model. The model in figure 1 will be used as a dynamic tool to aid this synthesis, and refined through an iterative process. Each pathway on the Eye Health—SDG logic model will then be examined separately and relevant evidence for that pathway will be collated and summarised, including measures of effectiveness where available. The extent of evidence supporting each pathway on the logic model will be indicated, for example, by differing font styles or typographical emphasis (eg, italics).[6] If sufficient homogeneous studies are found on a particular SDG-related outcome we will consider meta-analysis, and will develop a detailed protocol for this separately.

## Patient and public involvement statement

This protocol was developed with input from the Commissioners of the Lancet Global Health Commission on Global Eye Health, which includes a diverse internationally representative group of people with lived experience

of vision impairment, policy makers, academics, clinicians, government eye health programme leaders and advocacy specialists (see Acknowledgements section for full list of involved Commissioners).

## ETHICS AND DISSEMINATION

Ethics approval is not required for this review, as it will only include published data. The results of this review will be used to develop a framework for how eye health contributes to the advancement of the SDGs. We will publish our findings in an open-access, peer-reviewed journal and develop an accessible summary of the results for website posting and stakeholder meetings. A summary of the results will also be included in the ongoing Lancet Global Health Commission on Global Eye Health.[5] We anticipate that the findings of this work will be of considerable interest to multiple stakeholders: people with lived experience of vision impairment, eye health professionals, clinicians, policy makers and the development community. We expect the information will be useful for policy debate and advocacy in the wider development community. It will also guide eye health researchers to where there are gaps in evidence and identify areas for future research.

**Author affiliations**

[1]London School of Hygiene and Tropical Medicine, International Centre for Eye Health, London, UK

[2]Manchester Royal Eye Hospital, Manchester, UK

[3]School of Optometry and Vision Science, The University of Auckland, Auckland, New Zealand

[4]Department of Clinical Medicine, Kenya Medical Training College, Nairobi, Kenya

[5]Division of Ophthalmology, Universidade de São Paulo Faculdade de Medicina de Ribeirão Preto, Ribeirão Preto, Brazil

[6]Sight Savers International, Islamabad, Pakistan

[7]Department of Ophthalmology, Mbagathi Hospital, Nairobi, Kenya

[8]School of Psychological and Cognitive Sciences, Peking University, Beijing, China

[9]Centre for Public Health, Queen's University Belfast, Belfast, UK

[10]Moorfields Eye Hospital, London, UK

**Acknowledgements** We acknowledge colleagues and Commissioners of the Lancet Global Health Commission on Global Eye Health who provided suggested links between eye health and the SDGs during the model development process (Brandon Ah Tong, Andrew Bastawrous, John Buchan, Rènée du Toit, Hannah Faal, Michael Gichangi, Clare Gilbert, Jost Jonas, Hannah Kuper, Fatima Kyari, Van Charles Lansingh, Ana Patrícia Marques, Ciku Mathenge, GVS Murthy, Thulasiraj Ravilla, Juan Carlos Silva, Anthony Solomon, Bonnielin Swenor, Hugh Taylor, Aubrey Webson).

**Contributors** MJB conceived the idea for the review. JHZ, JR, MJB and NC developed the logic model. JHZ and JR drafted and revised the protocol with suggestions from MJB, NC, NM, CB, SY, JF, CO, CJ who reviewed the protocol and provided feedback on the draft. IG constructed the search.

**Funding** MJB is supported by the Wellcome Trust (207472/Z/17/Z). JR is a Commonwealth Rutherford Fellow, funded by the UK government through the Commonwealth Scholarship Commission in the UK. The Lancet Global Health Commission on Global Eye Health is supported by The Queen Elizabeth Diamond Jubilee Trust, Moorfields Eye Charity (grant number GR001061), NIHR Moorfields Biomedical Research Centre, The Wellcome Trust, Sightsavers, The Fred Hollows Foundation, The SEVA Foundation, The British Council for the Prevention of Blindness and Christian Blind Mission.

**Competing interests** None declared.

**Patient and public involvement** Patients and/or the public were not involved in the design, or conduct, or reporting, or dissemination plans of this research.

**Patient consent for publication** Not required.

**Provenance and peer review** Not commissioned; externally peer reviewed.

**ORCID iDs**

Justine H Zhang http://orcid.org/0000-0001-8385-2003

Jacqueline Ramke http://orcid.org/0000-0002-5764-1306

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
