## [Reviewer comments · BMJ Open]

ARTICLE DETAILS

TITLE (PROVISIONAL)	Global Eye Health and the Sustainable Development Goals: Protocol for a Scoping Review
AUTHORS	Zhang, Justine; Ramke, Jacqueline; Mwangi, Nyawira; Furtado, João; Yasmin, Sumrana; Bascaran, Covadonga; Ogundo, Cynthia; Jan, Catherine; Gordon, Iris; Congdon, Nathan; Burton, Matthew J

VERSION 1 – REVIEW

REVIEWER	Kahlia McCausland Curtin University, Western Australia
REVIEW RETURNED	10-Dec-2019

GENERAL COMMENTS	Please find some comments and suggestions for consideration in the attached file. My comments are really queries for further detail which will aid readers understanding of your rationale and methods. It appears you have addressed all the required content for a scoping review and the manuscript is well written, thank you. - The reviewer provided a marked copy with additional comments. Please contact the publisher for full details.
---

REVIEWER	Tanvir Chowdhury University of Calgary, Canada
REVIEW RETURNED	22-Dec-2019

GENERAL COMMENTS	In the current review protocol, the authors aim to summarize the nature and extent of the published papers that demonstrates a link between improved eye health and advancement of the Sustainable Development Goals (SDGs). There are few suggestions that will help the readers to comprehend the proposal more easily. (1) A table or text description for the search-terms needs to be provided in the manuscript. Though the authors have provided the Medline search terms in appendix (Appendix 3: MEDLINE search terms), but that are not easily comprehensible for readers who are not that much familiar with executing search. (2) A discussion section is essential. The section needs to contextualize the importance, impact, and outcome of the proposed review. Also, limitations of the study need to be mentioned in the discussion section. It might be helpful for the authors if they go through some recent scoping review protocols published in the BMJ Open.
---

	(3) Though the authors has mentioned, “As we plan to review existing published literature only, this scoping review will be performed without specific patient involvement” – I will suggest the researchers to push the envelope little bit by getting community member / citizen researcher / policy maker involved in their study proposal. Their input will also be helpful for formulating the overall study as well as make sure the potential impact have meaningfulness to the end-users.
--	--

VERSION 1 – AUTHOR RESPONSE

Reviewer 1:

Please find some comments and suggestions for consideration in the attached file. My comments are really queries for further detail which will aid readers understanding of your rationale and methods. It appears you have addressed all the required content for a scoping review and the manuscript is well written, thank you.

RESPONSE: Thank you very much for your helpful comments and suggestions – we have amended our manuscript in response to your comments (track changes throughout). We have used the terminology outlined in PRISMA-ScR, so retained ‘data items’ instead of switching to ‘data extraction’.

For ease of review we have copied the Reviewer’s comments from the attached file into the list below, identifying them by Page and line Number as they appeared on the version sent back to us by the editorial office.

Abstract

Page 2, Line 9 “...will summarise?”

RESPONSE: This change has been made.

Introduction

Page 4, Line 5 “...around the world...”

RESPONSE: This change has been made.

Page 4, Line 11 “Suggest more background is provided before leaping straight into the review. Further on it is identified that SDG 3 is in relation to improved eye health. Can this further be fleshed out, and why”

RESPONSE: We have expanded the background on page 4 about the SDGs in the first paragraph and added a point of clarification around SDG3 in paragraph 3.

Methods and Analysis

Page 6, Line 28 “Does this include other peer reviewed articles other than primary research studies?”

RESPONSE: We have provided additional clarification to the eligibility section. We will include meta-analyses as well as primary research studies.

Page 6, Line 41 “Using any reference management software to aid this process?”

RESPONSE: The “Search” paragraph has been revised on page 6. We are using EndNote and Covidence.

Page 6, Line 58 “On how many studies? Do you have any sense of what might be included in the initial data chart? You can then report in your results manuscript how this may have changed through the iterative”

RESPONSE: Not until we do the work.

Page 7, Line 8 “Is the information listed here different to what will be included in the "data charting process"? This is unclear”

RESPONSE: Clarified in the data charting paragraph on page 7.

Page 7, Line 54 “Unsure what different text types means”

RESPONSE: This has been clarified top of page 8 – different font styles / emphasis.

Reviewer 2:

In the current review protocol, the authors aim to summarize the nature and extent of the published papers that demonstrates a link between improved eye health and advancement of the Sustainable Development Goals (SDGs).

There are few suggestions that will help the readers to comprehend the proposal more easily.

(1) A table or text description for the search-terms needs to be provided in the manuscript. Though the authors have provided the Medline search terms in appendix (Appendix 3: MEDLINE search terms), but that are not easily comprehensible for readers who are not that much familiar with executing search.

RESPONSE: Thank you for your suggestion – we have worked with our Information Specialist from Cochrane Eyes and Vision to add a text description of the search terms in the Methods section (under the section entitled ‘Search’ on page 6) – we hope this helps clarify the execution of the search for readers.

(2) A discussion section is essential. The section needs to contextualize the importance, impact, and outcome of the proposed review. Also, limitations of the study need to be mentioned in the discussion section. It might be helpful for the authors if they go through some recent scoping review protocols published in the BMJ Open.

RESPONSE: Many thanks for these suggestions – we have looked at other scoping review protocols published by BMJ Open, and expanded our ethics and dissemination section into a short discussion section contextualising the impact that we anticipate this scoping review will have. We have discussed the limitations of the study in the ‘Strengths and Limitations’ section (according to BMJ Open guidance).

(3) Though the authors has mentioned, “As we plan to review existing published literature only, this scoping review will be performed without specific patient involvement” – I will suggest the researchers to push the envelope little bit by getting community member / citizen researcher / policy maker involved in their study proposal. Their input will also be helpful for formulating the overall study as well as make sure the potential impact have meaningfulness to the end-users.

RESPONSE: This is a very good suggestion for clarification. We have already involved policy makers, ambassadors for people living with disabilities, and several people with lived experience of visual impairment, who make up our Lancet Commissioners Group. Our Commissioners have been involved in the development of the protocol – we had acknowledged this in our ‘Acknowledgements’ section, and we have now amended the ‘Patient and Public Involvement Statement’ to reflect the involvement of the Commissioners here as well.

VERSION 2 – REVIEW

REVIEWER	Kahlia McCausland School of Public Health, Curtin University, Australia
REVIEW RETURNED	30-Jan-2020
GENERAL COMMENTS	Thank you for the opportunity to review the revised manuscript. It reads very well and appears all relevant areas of the PRISMA checklist have been covered. I do not have any further comments or suggestions.
REVIEWER	Tanvir C Turin University of Calgary

REVIEW RETURNED	14-Jan-2020
-------------

GENERAL COMMENTS	None
------